

# Characterization of AOD anomalies in September and October 2022 over Skukuza in South Africa

Marion Ranaivombola[1], Nelson Bègue[1], Farahnaz Fazel-Rastgar[2], Venkataraman Sivakumar[2,3], Gisèle Krysztofiak[4], Gwenaël Berthet[4], Fabrice Jegou[4], Stuart Piketh[5], and Hassan Bencherif[1,2]

[1]Laboratoire de l'Atmosphère et des Cyclones, UMR 8105 CNRS, Université de la Réunion, Reunion Island, France
[2]School of Chemistry and Physics, University of KwaZulu Natal, Durban 4000, South Africa
[3]National Institute for Theoretical and Computational Sciences, University of KwaZulu Natal, Durban 4000, South Africa
[4]Laboratoire de Physique et Chimie de l'Environnement et de l'Espace (LPC2E), Université d'Orléans, 45100 Orléans, France.
[5]Unit for Environmental Science and Management, North-West University, Potchefstroom 2520, South Africa

**Correspondence:** Marion Ranaivombola (marion.ranaivombola@univ-reunion.fr)

**Abstract.** This case study presents the evolution of aerosol optical properties during Intensive Observational Period (IOP) of the Biomass Burning Aerosol Campaign (BiBAC) in the Kruger National Park at Skukuza, between 18 and 23 September (Event 1) and 9 and 17 October (Event 2) 2022. The aerosol classification from sun-photometer data is consistent with the CALIOP, showing a predominance of biomass burning aerosols. The transport of CO and aerosols shows a southeastward transport over Southern Africa and toward the SWIO basin. The vertical distribution of aerosols spans vertically from the surface to 6 km (Event 1) and until 10 km (Event 2). The study discusses the synoptic conditions that have favored the significant perturbation of aerosol loading from September to October 2022. During Event 1, the "river of smoke" phenomenon was driven by surface ridge tongues with the westerly wave not being converted into a COL. During Event 2, a surface heat low, mid-tropospheric anti-cyclonic system, and warm air column resulted in stable conditions, which was also influenced by strong subsidence. The study investigates the long-range transport of biomass burning from South America to Southern Africa, with the plume exiting over southern Brazil, likely driven by the Southern American low-level jet (SALLJ), which is driven by climate forcings like El Niño–Southern Oscillation (ENSO) and Madden-Julian Oscillation (MJO). Further research addresses to the contribution of biomass burning plumes from Southern Africa and South America to those observed during both events and determines the atmospheric pattern.

## 1 Introduction

Atmospheric aerosols influence the global climate system by scattering and absorbing incoming solar radiation, changing cloud microphysics, and through their effect on radiative forcing (Albrecht, 1989; Twomey, 1977; Feingold and Siebert, 2009). Assessing climate change relies on the understanding of the processes controlling the burden and distribution of aerosols. This is more crucial in the Southern Hemisphere, due to the low number of observations in comparison with the Northern Hemisphere. The Southern American and Southern African regions are recognized to be significant primary sources of carbonaceous aerosols in the Southern Hemisphere through the biomass burning activity which occurs seasonally from June to November



(Randerson et al., 2012; van der Werf et al., 2017). This intense biomass burning activity has the potential to release a large number of pollutants, such as carbon monoxide and aerosols. These pollutants are harmful to both the environment and human health (Duflot et al., 2010, 2022; Holanda et al., 2020; Torres et al., 2007). Based on sun–photometer observations at the Reunion site, Duflot et al. (2022) showed that the AOD variability is mainly modulated by the biomass burning activity over Southern Africa and South America.

The low number of observations in Southern Africa during biomass burning season has led to achievement of several measurement campaigns in the 1990s, such as the Southern Tropical Atlantic Region Experiment in 1992 (STARE-92, Andreae et al., 1996), regrouping two scientific experiments: the Transport and Atmospheric Chemistry near the Equator-Atlantic (TRACE-A, Fishman et al., 1996) and the Southern Africa Fire Atmosphere Research Initiative (SAFARI-92, Lindesay et al., 1996). The TRACE-A experiment aimed to investigate the transport and chemistry of trace gases over Southern Africa and Equatorial Atlantic Ocean. At the same period, the SAFARI-92 was undertaken and focused on studying the impacts of savanna fires and biomass burning on the atmosphere, air quality, and climate. These experiments showed that biomass burning plumes are transported from both South America and Southern Africa over the Atlantic Ocean. During the biomass burning season, a photochemical reaction occurs between the associated emissions and solar radiation, which increases the concentration of ozone and leads to its accumulation in the Southern Hemisphere, from eastern Brazil to Australia (Fishman et al., 1996; Pickering et al., 1996; Thompson et al., 1996; Lindesay et al., 1996). SAFARI-2000 expanded upon SAFARI-92's research, encompassing a broader range of topics, including aerosols, clouds, and land-atmosphere interactions in Southern Africa (Swap et al., 2003). During SARAFI-2000, Schmid et al. (2003) observed a thick and widespread layer of aerosols that covered southern Africa, forming a smoke plume that extended from southeastern southern Africa to the Indian Ocean. Recently, the AEROCLO-sA (Aerosol, Radiation, and Clouds in Southern Africa, Formenti et al., 2019) campaign was carried out to study aerosols and their interactions with clouds and fog and biomass burning plumes transport. Based on data collected during this campaign, Chazette et al. (2019) and Flamant et al. (2022) investigated the interaction between tropical temperate troughs and cut-off lows that form over the west coast. These phenomena potentially contribute to the transport of biomass burning aerosols, also known as "rivers of smoke", from fire-prone regions in the tropical band to temperate mid-latitudes. River of smoke occurrence is generally due to a direct eastward transport of biomass burning plume from the continent toward the southwestern of Indian Ocean (SWIO) basin. This eastward transport is one of the five trajectories that govern the air flows over Southern Africa (Garstang et al., 1996). These trajectories lead to the transport of pollutants toward the SWIO basin (Clain et al., 2009; Duflot et al., 2010) and the southeast of Australia (Pak et al., 2003). Recently, Gaetani et al. (2021) performed a classification of the synoptic variability over 14 years (2003—2017) that controls the spatial distribution of biomass burning aerosol in Southern Africa and South Atlantic Ocean. They reported that the synoptic variability is composed of six weather regimes. Four of them correspond to disturbances traveling at mid-latitudes and the two-remaining account for pressure anomalies in the South Atlantic. These mid-latitude anomalies patterns may act in the formation of river of smoke and are consistent with Garstang et al. (1996).

The vertical distribution of aerosols determines their radiative impact, as well as their atmospheric residence time, which will affect any aging processes and the resultant horizontal distribution following advection (Darbyshire et al., 2019; Morgan





et al., 2020). An overview (2008-2021) of the spatiotemporal variability of aerosols in Southern Africa and Reunion Island was recently investigated by Ranaivombola et al. (2023), using a combination of sun–photometer and satellite observations. They highlighted that vertical profiles of elevated smoke reach 3-4 km when the biomass burning is at its maximum. Using the

HYSPLIT back-trajectory model, they also demonstrated that intercontinental transport from South America and the Atlantic Ocean to southern Africa may rise during the biomass burning season. The quantification of the influence of biomass burning aerosols on the climate system by numerical models is very dependent on how accurate the parameterization of aerosols is represented. However, for the tropical biomass burning regions, aerosol vertical distributions and their driving dynamical processes are poorly documented. Furthermore, the effort of measurements over Southern Africa during the aforementioned

campaigns (SAFARI-92, SAFARI-2000 and AEROCLO-sA) were mainly performed over the western coast. Until now, few studies and measurement campaigns have been carried out on the vertical distribution of aerosols from biomass burning and their transport to the SWIO basin.

Given the lack of observations on aerosols on the east coast of Southern Africa and their transport over the SWIO basin, an effort has been done to perform regular aerosols and chemical measurements during the 2022 biomass burning season in the

framework of the Biomass Burning Aerosols Campaign (BiBAC). BiBAC is a multi-instrumental campaign for quantifying optical properties of aerosol over Southern Africa and Indian ocean, as well as the associated transport at regional and large scale. The intensive observational phase (IOP) took place in the Kruger National Park (KNP), at the Skukuza camp, from August to November 2022. Given KNP is highly exposed to the biomass burning plume, it is considered as a core base site to investigate the characteristics and impacts of biomass burning emissions over Southern Africa (Garstang et al., 1996;

Swap and Tyson, 1999). Furthermore, this site has been hosting a Cimel Sun–photometer for several decades. During the IOP period, an instrumental set including a mobile lidar, a ground-based and in situ optical particle counters (LOAC, Light Optical Aerosols Counter) and a mobile monitoring station for measurement of surface aerosol and trace gas chemical properties and concentrations have been deployed in the KNP.

This study reports on the analysis of the spatiotemporal and vertical evolution of aerosol optical properties during the

BiBAC campaign. In addition, it aims to provide a general description of the dynamic context that has impacted the air mass distributions observed during the campaign. In this work, the temporal, horizontal and vertical distributions of aerosols are determined from ground-based observations collected over Southern Africa as part of the BiBAC campaign and combined with satellite observations and reanalysis data. The paper is organized as follows: Section 2 describes the observations used in this study. An overview of sun–photometer observations during the IOP, as well as the extraction of aerosol loading events are

presented in Section 3. The transport of CO and aerosols from both ground-based and satellite observations is highlighted in Section 4. The discussion on the synoptic conditions that drive the transport aerosol plumes is provided in Section 5. A final concluding remarks of the study are given in Section 6.



## 2 Site Location, Instrumentation and Methods

### 2.1 Site Location

The main study site of BiBAC campaign is the N'RWASHWITSHAKA Research Camp, in the Kruger National Park (24.98°S, 31.60°E), near to the main tourist rest camp, Skukuza. This site is located in the Mpumalanga province, in the eastern part of South Africa and shares its borders with Mozambique and the provinces of Limpopo, Gauteng, and KwaZulu-Natal (see Figure 1). The sun–photometer (pink cross, Figure 1a) was located approximately 1.7 kilometers from the N'RWASHWITSHAKA Research Camp, where ground-based instruments were installed (see orange cross in Figure 1b). Ground-based remote sensing instruments were deployed from September to November 2022, collocated to the sun–photometer.

To follow the transport of the aerosol plume toward the SWIO basin, we also used the sun–photometer data at Maputo site (-25.95°S, 32.59°E). Maputo is located along the southeastern coast of Mozambique, near to the border with South Africa. The sun–photometer, operated by the University Eduardo Mondlane (UEM), is located close to the sea (purple cross, Figure 1a). The two sun–photometer stations are approximately 148 kilometers apart.

### 2.2 Instrumentations and methods

#### 2.2.1 Ground-Based observations: AERONET sun–photometer

The AERONET (AErosol RObotic NETwork) sun–photometer (model CE318) is a passive remote-sensing instrument that measures the optical and microphysical properties of vertically integrated aerosols in the atmosphere by analyzing the attenuation of solar radiation. Measurements are taken under cloud-free and daytime conditions, at wavelengths of 340, 380, 440, 500, 670, 870, 940, and 1020 nm and at 15-minute intervals. It is noteworthy that level 2 sun-photometer data was not available at the time of this study. We therefore used level 1.5 data (Version 3) to analyze direct sun measurements of aerosol optical depth (AOD) at 440 and 500 nm, Ångström Exponent (AE) at 440 and 870 nm, the aerosol volume size distribution (AVSD) in the 0.05–15 µm range, single-scattering albedo (SSA), extinction AE (EAE), fine mode fraction (FMF) derived from the aerosol size distribution data obtained by the sun–photometer. To prevent the substantial inversion errors caused by the restricted aerosol information content, the SSA almucantar retrievals were only available when AOD at 440 nm is greater than 0.4 (Dubovik and King, 2000; Smirnov et al., 2000). Eck et al. (1999) and Dubovik et al. (2006) estimated the uncertainty in AOD measurements under cloud-free conditions to be 0.01 in the visible and near-IR and increasing to 0.02 in the ultraviolet (340 and 380 nm). AVSD retrieval errors typically do not exceed 15-35% (depending on the aerosol type) for each particle radius bin within the 0.1-7.0 µm range. Errors for extremely small particles (0.05-0.1 µm) and very large particles (7-15 µm) for a given particle radius bin can be as high as 35-100% (Dubovik and King, 2000; Smirnov et al., 2000). The uncertainty in SSAs values was expected to be of the order of 0.03-0.05, depending on aerosol type and loading (Eck et al., 2003; Alam et al., 2011; Yu et al., 2016).

Several aerosol classification methods from sun–photometer measurements have been used in some studies over various environments in the world (Lee et al., 2010; Giles et al., 2012), such as South Africa (Kumar et al., 2014, 2017, 2020; Adesina



et al., 2017; Ranaivombola et al., 2023), India (Kaskaoutis et al., 2009; Patel et al., 2017), Buenos Aires (Cúneo et al., 2022)
and Australia (Yang et al., 2021). In this study, we classified aerosols by using the following parameters: AOD, AE, EAE, SSA,
and FMF, and a variety of threshold values (Figure 2) and all–point data to have the largest amount of data. We selected the
timestamps that were common to each pair of parameters with a 120–second interval.

The AOD-AE method allows to classify four aerosol types, using threshold values: biomass burning/urban industrial (BB/UR),
clean marine (CM), desert dust (DD) and mixed aerosols (MX), (Kaskaoutis et al., 2009; Patel et al., 2017; Giles et al., 2012).
This method was already used in several works over South Africa (Kumar et al., 2017, 2020; Ranaivombola et al., 2023). The
BB/UR aerosol types can be distinguished using SSA, in combination with the EAE size indicator. The SSA correspond to the
ratio of scattering efficiency to extinction and can be used to distinguish between absorbing (SSA<0.95) and non-absorbing
aerosols (Sinyuk et al., 2003; Lee et al., 2010). The EAE-SSA method is known to allow identifying three major aerosol types:
dust (DD), urban/industrial (UR), and biomass burning (BB). This method was used and reported in numerous studies and in
several countries like South-Africa, India, Bangladesh and Pakistan (Giles et al., 2012; Bibi et al., 2016; Adesina et al., 2017;
Kumar et al., 2020; Zaman et al., 2022). Note that in these references, urban/industrial is abbreviated to UI, instead of UR.
We have chosen to standardize, with BB/UR from the AOD-AE method, according to the work of Ranaivombola et al. (2023).
The combination of SSA with FMF parameter reveals the characteristics of the absorbing aerosols. FMF is used to determine
the dominant size mode, and its wavelength was interpolated from 500 to 550 nm using AE at 440–870 nm. Aerosols were
classified into four groups: dust (DD), non-absorbing aerosols (NA), absorbing aerosols (AA) and mixture (MX). The AA type
is divided into highly-absorbing (HA), moderately-absorbing (MA), and slightly-absorbing (SA) depending on the range of
SSAs (Figure 2). This method was used in several studies of Lee et al. (2010); Giles et al. (2012); Adesina et al. (2017); Boiyo
et al. (2019); Kumar et al. (2020) and Zaman et al. (2022).

The calculation of aerosol frequencies is based on the equation 2 in Ranaivombola et al. (2023), where $N_s$(type), is the num-
ber of occurrences for a given aerosol type and period (i.e., September-October-November (SON) 2022, September, October,
Event 1, and Event 2).

Sun–photometer data for Skukuza and Maputo ranges from 1998 to 2022 and from 2019 to 2022, respectively, although
Skukuza sun–photometer was out of operation from 2012 to March 2016 owing to calibration and maintenance issues (Adesina
and Piketh, 2016). As a result, data spans the years 1998 to 2011 and 2016 to 2020. This represents 4605 days of AOD and
AE measurements, out of which 246 were in 2022, while the Maputo sun–photometer has 868 daily observations, with 267 in
2022.

## 2.3    Satellite observations

### 2.3.1    MODIS-Aqua Aerosol observations

The Moderate Resolution Imaging Spectroradiometer (MODIS) is carried onboard NASA's Aqua (EOS PM-1) satellite,
which has helio-synchronous orbit and has been active since 2002. MODIS records data in 36 different spectral bands with
wavelengths ranging from 0.4 to 14.4 μm and spatial resolutions ranging from 250 m to 1 km depending on the chan-



nel. MODIS products give several parameters derived from the radiance measurement. These quantities are the distribution of cloud cover, the amount of water vapor in an atmospheric column, the temperature distribution, and the characteristics of aerosols. The data used in the present study is the "Combined Dark Target and Deep Blue AOD at 550 nm, for land and ocean (MOD08_D3_v6.1)", collected from the Aqua platform and downloaded from the Giovanni website (https://giovanni.gsfc.nasa.gov/giovanni/, last accessed on July 11, 2023).

Regarding fire data, the products "Thermal Anomalies/Fire locations 1km FIRMS V0061 (MCD14ML)" from Aqua platform were downloaded from the FIRMS platform (https://firms.modaps.eosdis.nasa.gov/, last accessed on November 7, 2023). A confidence class is attributed to each pixel-fire ("low", "nominal" or "high"). For this study we only used the "high" confidence class, where the confidence values range from 80 to 100% (Giglio, 2015).

### 2.3.2 CALIPSO Aerosol Profiles

The Cloud Aerosol Lidar Infrared Pathfinder Satellite Observations (CALIPSO) was launched in April 2006, in a joint mission between NASA and French Agency CNES, to investigate the effects of aerosols and clouds on the Earth's radiation budget and climate. It is a polar-orbiting satellite of the A-Train Constellation, flying at a height of 705 km with a 16-day cycle. The Cloud Aerosol Lidar with Orthogonal Polarization (CALIOP) is one of the three instruments of CALIPSO and is an elastic backscatter lidar, which operated at two wavelengths: 532 and 1064 nm. It uses two 532 nm receiver channels and a channel measuring the total 1064 nm return signal (Hunt et al., 2009). The dataset used in this study is the Level-2 Aerosol Profiles (V4-51), for a period covering the IOP and downloaded from the EarthData website (https://search.earthdata.nasa.gov/, last accessed on October 23, 2023). The V4-51 is an upgrade of V4-21 by improving the accuracy of the smoke layer above clouds and clearing clouds in the planetary boundary layer that are misclassified as aerosols (Getzewich, 2023). Data were selected on a box with latitude range from 10°S to 40°S and longitude from 10°E to 40°E (Figure 1c). We used aerosol extinction profiles at 532 nm, as well as the layer classification information (i.e., aerosol and "clean-air", meaning air without aerosols). Aerosol types given by the CALIOP algorithm are clean marine, dust, polluted continental/smoke, clean continental, polluted dust, elevated smoke, and dusty marine. The quality-filtering procedure of data is described in Tackett et al. (2018). We have adapted them for Level–2, since Tackett et al. (2018) used these criteria for Level–3 data by including (1) -100 $\leq$ cloud aerosol discrimination score $\leq$ –50, (2) extinction quality control flag = 0 or 1, (3) extinction uncertainty > 99.99 $\mathrm{km}^{-1}$ and (4) extinction for "clear air" = 0 $\mathrm{km}^{-1}$. Average profiles of total aerosol extinction at 532 nm were obtained by the same averaging method described in section 4.3 by Tackett et al. (2018). To obtain the profiles for each type, one applies a mask for each type using the layer classification information.

### 2.3.3 MetOp-b IASI CO measurements

The Infrared Atmospheric Sounding Interferometer (IASI) on board the Meteorological Operational (MetOp) satellite uses a Fourier transform spectrometer to measure the infrared radiation at wavelengths ranging from 6.62 to 15.5 μm during daytime and nighttime (Clerbaux et al., 2009; Coheur et al., 2009; Clarisse et al., 2011). It was designed for global observations of chemical species (such as ozone, sulfur dioxide, and carbon monoxide) with a vertical range covering the troposphere and the



lower stratosphere. In the present study we used the daily IASI/MetOp-B carbon monoxide (CO) dataset, which is processed using the Fast Optimal Retrievals on Layers for IASI (FORLI) software (Hurtmans et al., 2012) and downloaded from the AERIS platform (https://iasi.aeris-data.fr/co, last accessed on December 8, 2023).

### 2.4 CAMS global reanalysis (EAC4)

The Copernicus Atmosphere Monitoring Service (CAMS) provides global reanalysis (EAC4) dataset of atmospheric composition that covers 2003 to 2022 (Inness et al., 2019). This dataset was built on the knowledge gathered during the previous Monitoring Atmospheric Composition and Climate (MACC) reanalysis and the CAMS interim reanalysis. The CAMS reanalysis uses the ECMWF's Integrated Forecasting System (IFS) model to retrieve total column of carbon monoxide (CO); tropospheric column of nitrogen dioxide (NO2); aerosol optical depth (AOD); and total column and partial column, and pro-

files of ozone (O3) from satellites measurements (e.g. IASI, MOPITT and TROPOMI for the CO and MODIS and VIIRS for the AOD). The reanalysis has a high horizontal resolution of $0.75° \times 0.75°$ and provides more chemical species at a high temporal resolution (3-hourly analysis fields) than the previous CAMS interim reanalysis. In the present study, we used CAMS reanalysis at 12:00 PM to follow-up with the synoptic situation during biomass burning aerosol events detected during the IOP. We used the mean sea level pressure (mslp in hPa), the AOD at 550 nm, the total column of CO $(\text{molecules.cm}^{-2})$, the geopo-

tential $(\text{m}^2.\text{s}^{-2})$ and vertical velocity $(\text{Pa.s}^{-1})$ at the 700 hPa pressure level. The geopotential height (in meters) is obtained by dividing the geopotential by the Earth's gravitational acceleration: $9.80665 \text{ m.s}^{-2}$, which is a fixed value in the IFS model.

This dataset has already been used in Flamant et al. (2022) to describe the synoptic conditions over Southern Africa in 2017. Reanalysis data were downloaded from the Atmospheric Data Store (ADS) website (https://ads.atmosphere.copernicus.eu/, last accessed on December 19, 2023).

## 3   Evolution of aerosol properties during BiBAC

Figure 3a displays AOD at 500 nm (green line) and AE at 440–870 nm (purple line) recorded at Skukuza during the IOP. Daily average AOD and AE values in September and October 2022 are compared with a multiyear monthly average of the entire dataset (September and October 1998-2011 to 2016-2021). Monthly average values of AOD and AE lie in the 0.2-0.3 and 1.2-1.5 ranges, respectively, and reveal a decrease with time, suggesting a slowdown in biomass burning activity between

early September and late October. These values are similar to those reported by Ranaivombola et al. (2023) during the austral spring (September to November). The daily AOD and AE values show the same trend as the multiyear monthly average values, except during the 3rd week of September (from 18 to 23) and during the 2nd week of October (from 9 to 17), where AOD and AE values increased and reached maximums of 0.7 and 1.9, respectively. Unfortunately, the photometers did not work on 20 and 21 September due to the rainy conditions.

To characterize the anomalies in aerosol loading over the study site, the quartiles *Q1*, *Q2* and *Q3* of the relative difference of the AOD (AOD$_{RD}$) were computed and correspond to the following equation:



$$AOD_{RD} = 100 \times \frac{AOD_{2022} - AOD_{mm}}{AOD_{mm}} \qquad (1)$$

where $AOD_{mm}$ representing the multiyear monthly average AOD values (Figure 3b). At least two values of $AOD_{RD}$ exceeding the *Q3* line within a given period are considered high aerosol load events at ±1 day. Two aerosol events were thus identified
during the IOP: Event 1, from 18 to 23 September, and Event 2, from 9 to 17 October, highlighted with yellow and blue shaded areas respectively in figure 3b.

In the following part, the 2 events are characterized in terms of different aerosol classification types. Figure 4 shows the aerosol type occurrence frequencies at Skukuza for September (dark blue dots) including Event 1 (light blue dots), and for October (brown dots) including Event 2 (yellow dots), respectively. Grey dots represent the austral spring 2022 period (called
SON hereafter, for the September-October-November period). As mentioned above, the aerosol classification is carried out using three methods, on scatter plot between each parameter of: AOD-AE (Figure 4a and b), EAE-SSA (Figure 4c and d), and FMF-SSA (Figure 4e and f).

Based on AOD–AE method, Skukuza exhibits higher frequencies for biomass burning/urban industrial (BB/UR) type (85%) than for mixed (MX) type (14%) during SON (Figure 4a and b). The other types are absent on both months. These values
agree with those obtained by Ranaivombola et al. (2023) during austral spring. In complement to this first classification, the EAE-SSA method allows to complete the distinction between Biomass burning (BB) and urban/industrial (UR) aerosols. BB aerosols predominate against UR type at Skukuza during SON period (61% against 30%, Figure 4c and d). An undetermined category (labeled n/a) corresponding to aerosol which does not fit in any category represents 8%. The FMF–SSA method allows to have more information on the absorbing and non-absorbing aerosols. The AA type (i.e., absorbing aerosols including
slightly, moderately and highly absorbing SA, MA, and HA) prevails during SON period, with frequencies of 98.3% (Figure 4e and f). In this category, aerosol is moderately and slightly absorbing, where MA and SA types account for 63.6% and 33%, respectively. The results found at Skukuza with this latter method are consistent with those reported by Lee et al. (2010). The same maximum frequencies are observed in the September and October period. The BB/UR and MX types appear to be predominating, with values of up to 80% and 2–12% respectively (Figure 4a and b). BB types predominate at Skukuza in
comparison with UR (58–60 against 30%, Figure 4c and d). AA type represents the highest frequencies, with values of 98%, where MA prevails with frequencies from 57 to 63% (Figure 4e and f).

Looking at Events 1 and 2, BB/UR predominates at 100% for both events (Figure 4a and b). Based on EAE-SSA method, BB type prevails at Skukuza for both events with frequencies from 59–85% (Figure 4c and d). With the FMF–SSA method, AA type (MA and SA) is predominant for both events, with frequencies from 99.5 to 100% (Figure 4e and f).
To summarize the aerosol classification over Skukuza with the three methods, BB/UR aerosols type prevails in front of other types during the IOP and as well during the two events (see table 1). Using the SSA with other inversion parameters, we have determined that BB aerosol type prevails, and aerosols are moderately absorbing.



With the Sun–photometer, we have determined two anomalies of AOD with aerosols type that are mainly BB aerosols linked to biomass burning over Southern Africa. In the following section, the transport and the vertical distribution of biomass burning plume is investigated for both events.


## 4 Transport of CO and aerosol plume

In order to characterize air masses transport during the detected aerosol events over the Skukuza site from sun–photometer measurements, additional data from the neighboring site of Maputo and from satellite records are considered and presented in this section.

Figure 5 represents the daily aerosol volume size distribution (ASVD) derived from sun–photometer covering each event at Skukuza (a and b) and Maputo (c and d), for September (upper plots) and October (lower plots). Here daily measurements refer to an average of all available observations per day. The daily ASVD are compared with the monthly multiyear average from 1998–2011/2016–2021 and 2019–2021, respectively for both sites.

The monthly multiyear AVSD have a bimodal shape with maximum at 0.03-0.04 $\mu m^{-3}/\mu m^{-2}$ for both sites in September
and October. They show a predominance of fine aerosol type at Skukuza (with radius 0.11- 0.14 $\mu m$ for both month) and a predominance of coarse aerosol type at Maputo (with radius of 3.85 $\mu m$) during the two months (Figure 5). Despite the limited number of observations at Maputo site, compared to Skukuza, it should be noted that the AVSDs of the fine mode observed in Maputo in 2022. The AVSD values in 2022 in the fine mode are about 4 times higher than those of the multi-year averages. The monthly multiyear AVSD values obtained at Skukuza are consistent with previous work over the site (Queface et al., 2011;
Adesina et al., 2017).

Daily AVSD obtained during Event 1 seem to be similar at both sites, at Skukuza and Maputo, which suggests that the 2 sites are influenced by similar air masses. During Event 2, the obtained AVSD profiles seem to differ from one site to another, especially for the coarse mode. AVSD on September 19 and 22 are greater than the multi-year AVSD values. The radius of the AVSD peak on September 19 is consistent with the one obtained by Eck et al. (2003) at Zambia during the 2000 biomass
burning season. One can note a change of radius between these two dates and for both sites, with a decrease of the AVSD on September 22. AVSD for coarse particles on October 15 at Maputo show also greater values than the multi-year AVSD values. This change of radius is associated with a decrease in aerosol concentrations. This could be relied on by the coagulation processes of smaller particles (Seinfeld and Pandis, 2016) or to the presence of two air masses with different aerosol types. Regarding aerosol types day by day, there is a change of aerosol types from September 19 to 22 (no shown). Aerosols were BB
and MA on September 19, while it change to UR and SA. However on October 15, the EAE-SSA method show undetermined aerosol and SA aerosols.

The CO is generated during the incomplete combustion of carbon compounds. It is one of the main pollutants emitted into the atmosphere by biomass burning activity. Given its lifetime (from weeks to months), it is often used as a tracer of biomass burning plume transport (Bencherif et al., 2020). In addition to aerosol data from MODIS (AOD), this study uses CO
measurements observed by IASI to monitor aerosol plumes.





Figure 6 displays averaged maps of total column of CO and AOD at 550 nm during the two events (panels a and b for Event1, panels c and d for Event 2), based on satellite observations from IASI and MODIS-Aqua. CO maps show high values over Southern Africa, mainly over the latitude band -5° to -25°, as well as the center of Brazil (Figure 6a and c). The CO plume spreads across Southern Africa west coast, reaching the Atlantic Ocean and the eastern part of Southern Africa. The center

of Brazil also has a CO plume that extends to the Atlantic Ocean. From CO maps, it evident that the plumes mixing from Southern Africa and South America takes place in CO values over the Atlantic Ocean. The CO plume propagates thereafter toward the SWIO basin over the eastern part of Southern Africa. AOD maps show similar structure between aerosol and CO plumes (Figure 6b and d). The aerosol plumes spread over the Atlantic Ocean, over the SWIO basin and from the center of Brazil to the Atlantic Ocean. The similarities between the structures of the CO and AOD plumes suggest that the AOD values

are the result of biomass burning. Although the main structures are the same, there are differences between the areas of CO and AOD maxima. High CO concentrations are observed in the latitude band from 0° to -10° over the Atlantic, which is not the case for AOD. The maximum of AOD over the western coast of Southern Africa seems to correspond to CO minima and/or missing CO data.

To better understand the horizontal transport and vertical distribution of the aerosol plume, we have used AOD from CAMS

Reanalysis and CALIPSO observations. CAMS Reanalysis have been used every 24 hours, at midday (see Figure 7 and Figure 10 in the following sub-sections) and every 3-h (see the animations 1 and 2, at the respective url: https://av.tib.eu/media/67052 and https://av.tib.eu/media/67051) for both events. CALIPSO overpasses have been selected over the Mozambique Channel to characterize aerosol vertical distribution during their transport to the SWIO basin.

## 4.1 Aerosol horizontal and vertical distribution during Event 1: September 18 to 23

Figure 7 represents the daily map of AOD (shading color) and mean sea level pressure (mslp, blue isoline) from CAMS reanalysis during Event 1. On September 18 (Figure 7), an aerosol plume with moderate (0.6–0.8) to high AOD values (0.8–1.2) was observed over the western coast and northern part of Angola. The plume then moved southeastward, passing over Namibia and reaching eastern South Africa. The Eastern Cape and Kwazulu-Natal provinces in South Africa had lower AOD values. The aerosol plume then propagated towards the SWIO basin, with values ranging from 0.6 to 1. On September 19, the aerosol

plume continues to propagate from north to south (Figure 7b). The aerosol plume shifts to eastern Southern Africa, affecting the eastern coast of South Africa and later Mozambique with AOD values greater than previous days and above one. On September 20, AOD values increase over Zimbabwe and Mozambique and spread toward the SWIO basin (Figure 7c). This suggests an intensification of biomass burning activity over the region at this date. The augmentation of AOD values over Zimbabwe and Mozambique was persistent during the day until September 21 at 00:00 UTC (see animation 1 and Figure 7d), where the

aerosol plume appears like a tongue from Angola to Mozambique with AOD values above 1. However, AOD values are lower (below 0.2) over South Africa, Namibia, and Botswana AOD values. The aerosol plume over Mozambique, spreading toward the SWIO basin, on September 22 and 23 (Figure 7e and f) depicts lower values than prior days (0.4–0.6). This suggests a decrease or a change of the intensity of biomass burning activity near Mozambique (see animation 1).



To characterize the aerosol vertical distribution toward SWIO, one selected CALIPSO flight over Mozambique Channel for both Events. Figure 8 represents CALIOP observations combined with CAMS reanalysis, respectively on September 18 at 23:55 UTC. The panel (a) of each figure represents the AOD map distribution at 550 nm derived from CAMS reanalysis (shading color), with the CALIPSO overpass orbit superimposed with blue line. The CALIOP cross-sections (latitude vs altitude) of extinction coefficient and aerosol type are given in panel (b) and (d), respectively. Averaged profiles of the extinction coefficient (total and per aerosol type) at 532 nm are displayed on panel (c). Each profile is averaged along the latitude represented on panel (a), see section method for details. The CALIPSO orbit on September 18 crosses the aerosol plume over the Mozambique channel in a latitude band from -17° to -35° (Figure 8a). This latitude band depicts two aerosols layers with extinction coefficient values from 0.08 to 0.2 $km^{-1}$. One from the surface to 3 km (-17° to -25°) and the other one from 2 to 6 km, (Figure 8b). Main aerosol type is this latitude band is elevated smoke, mixed with clean-marine and dusty–marine in the boundary layer (Figure 8c). The average profile reveals that the maximum of the extinction coefficient of elevated smoke is around 3 km (Figure 8d). The other CALIPSO flight over the Mozambique channel (September 20 at 12:37 UTC, 22 at 00:09 UTC and 23 at 23:44 UTC) reveals the same aerosols layer of elevated smoke from 4 to 6 km, in the latitude band from -30° to -20° of latitude (not shown).

The spatial variations of AOD values from CAMS reanalysis (Figure 7a to c) suggest an intensification of biomass burning activity from September 18 to 20. Figure 9 depicts the Fire Radiative Power (FRP) values from MODIS-Aqua from September 18 to 22. The biomass burning activity was in the latitude band -10° to -25° (Angola, Zambia, Zimbabwe, and Mozambique) and is associated with moderate values of FRP from 20 to 50 MW. Some locations show high FRP from 60 to 100 MW, such as the southeastern part of Angola or the central part of Mozambique on September 20. Regarding the FRP on September 18, there were dispersed over Mozambique over the north and south of the country, with low-energy fires (10 MW) to more energy fires (100 MW). FRP over Angola were ranging mainly from 60 to 100 MW and located in the south of the country (Figure 9a). Conversely, on September 20, FRP were mainly located in the center of Mozambique, with a high energy fire ranges from 60 to 100 MW and the number of fires decreased over south of Angola but not the fire intensity (Figure 9b). On September 22, FRP are located further east of Southern Africa compared to two days before (Figure 9c). The spatial variation of biomass burning activity could have impacted the AOD over Southern Africa. This could also explain the increase of aerosol radius observed at Skukuza and Maputo with sun–photometer (Figure 5a and c).

## 4.2 Aerosol horizontal and vertical distribution during Event 2: October 9 to 17

Figure 10 depicts AOD maps for Event 2, as previously mslp will be used in this section. On October 9, AOD maps and the animation 2 showed moderate to high AOD values (0.6–1) over Angola and its western coast, spreading towards the Atlantic Ocean. These moderate to high values are observed over Namibia and South Africa, while lower AOD values are found over Mozambique and Zimbabwe. The following days, from October 10 to 13, the aerosol plume exited by the western coast of Angola and expanded down the western coast of Southern Africa, spreading over South Africa and the SWIO basin (Figure 10b to e). As the aerosol plume spread along the western coast of Southern Africa, AOD values decrease over Botswana and Namibia. From October 14, the aerosol plume over the SWIO basin displayed higher values than those found above the



continent (Figure 10f). This large layer of aerosol was persistent until the event's end, matched with an increase in AOD values above Southern Africa from October 15 to the event's end (Figure 10g to i). According to CO and AOD maps the aerosol plume
propagated more broadly towards the SWIO basin than during Event 1 (Figure 6, Figure 7 and Figure 10).

Figure 11 depict the CALIPSO flight over Mozambique Channel, crossing the aerosol plume from -15° to -35° of latitude on October 13 at 00:02 UTC. The cross-sections of extinction coefficient and aerosol type reveal a large aerosol signature extending from -8° to -35° of latitude. The aerosol layer extends from the surface to 10 km and is organized into two parts. The first part (-35° to -20° of latitude) ranges from 1 to 10 km and consists mainly of elevated smoke with clean–marine aerosol in
the boundary layer. While the second part of the layer (-8° to -20° of latitude) is below 4 km. The first part is consistent with AOD map, with larger AOD values from 0.6 to 0.8, while the second part is out of range of the main AOD signal, with AOD values below 0.3. The other CALIPSO flight over the Mozambique channel (October 9 at 23:49 UTC, 16 at 00:15 and 17 at 23:51 UTC) and Southern Africa (October 14 at 00:39 UTC and 15 at 01:16 UTC) depicted an aerosol layer of elevated smoke for the surface to 10 km, in the -25° to -35° latitude band (not shown). Figure 12 shows the FRP values for MODIS-Aqua for
the Event 2. Such as Event 1, biomass burning activity was in the latitude band -10° to -25° with moderate to high values of FRP. During the event 2, fires tend to append eastward of Southern Africa, impacting mainly Zambia and Mozambique.

## 5  Discussion

From satellite observations and CAMS reanalysis, one can note that the two events were characterized by air mass transport highlighted by CO and aerosols distributions over the study region, in relationship with biomass burning activity in Southern
Africa. However, some differences in the horizontal and vertical structure of CO and aerosols can be pointed out. The shape of the plume toward the SWIO basin differs noticeably between Event 1 and Event 2. During Event 1, the plume crossed Mozambique and reached the SWIO basin, whereas during Event 2, the plume crossed a larger portion of Southern Africa's eastern coast, spanning Mozambique and South Africa. The variation in CO and AOD values over the Western Cape Province (South Africa) between events is noteworthy. During Event 1, AOD and CO concentrations are found to be low, 0.5 to 1
molecules.cm$^{-2}$ and 0.2, respectively, and they reached 2.5 molecules.cm$^{-2}$ and 0.4 to 0.5, respectively. More globally, during Event 2 these values are found to be higher. These difference can be linked to the number of fires. As show on the figure 12, the numbers of FRP seem to be greater during Event 2. The shape of CO and AOD plumes obtain during Event 1 looks likes the so-called phenomena of "river of smoke" reported in previous works during SAFARI-2000 on September 2, 2000 (Swap et al., 2003; Stein et al., 2003) and AEROCLO-sA campaign from September 2 to 6, 2017 (Chazette et al., 2019;
Flamant et al., 2022).

The vertical distribution of aerosols from CALIOP profiles are mainly composed of elevated smoke and span in altitude mainly from surface to  6 km during Event 1 and until 10 km during Event 2. The presence of elevated smoke is consistent with the BB aerosol type from the aerosol classification with sun–photometer presented in Section 3. The altitude range of elevated smoke is consistent with those reported by Ranaivombola et al. (2023).




In summary, the events observed during the BiBAC campaign are associated with the biomass burning activity in Southern Africa. Ground and satellite observations, as well as CAMS assimilation data, have highlighted the increase in aerosol and CO quantities over Southern Africa and across this region, towards the Atlantic Ocean and SWIO basin. However, the transport and outflow of aerosol plumes towards the Indian Ocean do not appear to occur in the same way during the 2 events. In the
following section, we discuss the synoptic conditions prevailing during these 2 events.

## 5.1 Synoptic conditions during Event 1

As presented in the previous section, the aerosol plume propagates southeastward over Southern Africa. Using the mslp on Figure 7, one can note that the transport of the aerosol plume toward the SWIO basin seems to be driven by the impact of the westerly wave and the high-pressure tongues which extended toward the continent. The movement takes place from Septem-
ber 18 and 19 and is well pronounced on September 20 and 21. In addition to the mslp, we have used vertical velocity and geopotential height at 700-hPa pressure level (Figure 13). The vertical velocity at pressure level (omega) allows to highlight the instability by vertical motion in the atmosphere. Negative values of omega is associated with upward motion, while positive values indicate downward motion. We have used the geopotential height at 700-hPa pressure level to follow wave propagation. The 700-hPa level corresponds approximately to the height of the maximum of aerosol profiles obtained from CALIOP
observations.

Figure 13 depicts the vertical velocity (shading color) and the geopotential height (blue isoline) at the 700-hPa pressure level during Event 1. The geopotential height pattern shows the movement of a westerly trough from September 18 to 19 over South Africa. The westerly wave is associated with a dynamic low–pressure system formed over the high latitudes which moves to the east (Figure 7) and causing instability over Southern Africa, with vertical velocity values of -0.2 $\mathrm{Pa.s^{-1}}$ on September
18 (Figure 13a). By the next day the trough has moved and located around central parts with higher negative omega ( - 0.7 $\mathrm{Pa.s^{-1}}$) in a head of the trough (Figure 13b). On September 20, the westerly flow has been separated from the main westerlies but did not form a closed cell. Also, it has not remained nearly stationary for several days for possible converting into a cut-off low. Based on the weather observation, there was rainfall reported (6.6 $\mathrm{mm}$ and 8.61 $\mathrm{mm}$ on September 20 and 21, respectively) over the study area showing weak frontal passage with very weak surface winds too. On September 22, the study
area impacted by anti-cyclonic ridge associated with rather stable condition. There is less instability behind the wave with downward motion (+0.4 $\mathrm{Pa.s^{-1}}$) while ahead there is upward motion (-0.6 $\mathrm{Pa.s^{-1}}$) but weaker than the previous day (Figure 13e). This is associated with surface high pressure tongue along with mid-tropospheric ridge, with less instability (omega at around 0.2 $\mathrm{Pa.s^{-1}}$). On September 23, a weak and minor trough impacted the study area (0.5mm rainfall reported).

Meteorological conditions during Event 1 have been influenced by the passage of the westerly wave favoring the eastern
transport corridor and consistent with the transport schemes established by Garstang et al. (1996). The influence of the westerly wave in the transport of aerosol is also consistent with the previous case studies of river of smoke characterized in the literature during SAFARI-2000 (Swap et al., 2003; Stein et al., 2003) and AEROCLO-sA (Flamant et al., 2022) campaigns. It is also consistent with the classification of synoptic variability over 14 years by Gaetani et al. (2021). They found that midlatitude pressure anomalies accounted for 60% of weather regime. These midlatitude anomalies are characterized by eastward travelling



disturbance of westerly flow influencing the formation of the river of smoke. This confirms the influence of the westerly wave on the formation of a river of smoke.

During SAFARI–2000, Stein et al. (2003) observed a link between westerly wave and the formation of a river of smoke happening during La Niña phase of El Niño–Southern Oscillation (ENSO). Rainfall conditions were characterized by a moderate La Niña phase, which started in austral summer 1998 (Roy et al., 2001), with sea surface temperatures between -1°

and -1.5°C. The Eline cyclone was the most contributor to intense growth of grass biomass and fire fuel for the 2000 dry season (Swap et al., 2003). In contrast to year 2000, La Niña phase was weaker in 2017 and 2022 with sea surface temperature values between -0.5° to -1°C. (see the Oceanic Niño Index (ONI) in NOAA website Climate Prediction Center, https://origin.cpc.ncep.noaa.gov/products/analysis_monitoring/ensostuff/ONI_v5.php). There was no weather phenomena similar to the Eline Cyclone in 2017 or 2022. However, one can note that the rivers of smoke reported in previous works and in

the current study occurred during La Niña phase of ENSO. As mentioned (Shikwambana et al., 2022) in the analysis of wildfires and associated emissions during ENSO phases in Southern Africa, ENSO phase has no direct influence on the biomass burning activity and their emissions but rather on rainy season and farming practices. The only difference with these previous case studies is that our Event 1 displays different synoptical conditions: the westerly wave was not converted to a cut-off low (COL). As synoptical conditions shown in Figure 13, there is no formation of closed cell that remains stationary for several

days, like in the works of Flamant et al. (2022). Based on these points, it would be interesting to make an overview of "river of smoke" and analyze their occurrence with ENSO phase, particularly in strong phase like 2010-2011.

## 5.2   Synoptic conditions during Event 2

In general, the mean sea level pressure maps analysis (figure 10) over the study area during October 9-17, 2022, displays an existence of the Angola heat-low (as a barotropic system) over the Southern Africa region. This is associated with an

upper subtropical ridge (figure 13) and warm air (isotherms are not presented here) over Angola, Botswana, and most parts of South Africa including the study area. Some parts in the western and southern areas of South Africa have been impacted by a baroclinic westerly system associated with the upper tropospheric westerly trough. However, the study area has been impacted by the mid-tropospheric subtropical anti-cyclonic system causing a stable condition. One can note that the south Atlantic high pressure (with two high–pressure centers displaced from Atlantic Ocean to the SWIO basin (Figure 10a). Both high–pressure

centers crossing over the southern part of the continent, from October 9 to 13 and from October 14 to 17, respectively. On one hand, the presence of these two high-pressure centers (figure 10a) may have limited the spread of the aerosol plume over continent and limited its outbreak into the SWIO basin and, on the other hand, the aerosol plume by passes the continent via the west coast of continent instead of crossing it, as during Event 1 (Figure 10a to e). The displacement of this cell explains the stagnation of the AOD and AE values over Skukuza obtained from the sun–photometer observations (Figure 3). This can also

explain the quasi–constant radius and concentration observed at Skukuza and Maputo (Figure 5b).

After the passage of the first high cell of pressure, the aerosol plume extends over a broader range toward the SWIO basin, which is consistent with MODIS observations (Figure 10d and e and Figure 6d). The location of the high-pressure cells on October 13 and 14, is like those observed on September 19 and 20 (Figure 10e and f). On October 14, the aerosol plume looks



like a river of smoke observed during Event 1. However, the high cell pressure passes further south and slower than during
Event 1, constraining the plume over the continent to propagate southward of the continent (Figure 10f to i).

As for Event 1, one used the vertical velocity and the geopotential height at 700–hPa pressure level (Figure 14). The figure shows the existence of the subtropical anti-cyclonic system accompanied with warm temperatures (isotherm are not represented here) dominated over the study area. The contour of geopotential height depicts the propagation of westerly wave south to the continent, affecting mainly the southwest coast areas in the South Africa. This figure displays no considerable upward motion,
resulting in stable conditions during event 2 over the study area. However, the longitude-pressure cross-section of the omega (not shown) over the study region shows rather higher absolute values (negative) mostly at levels of 800hPa ( - 0.4 $\mathrm{Pa.s}^{-1}$), 700hPa (with - 0.3 $\mathrm{Pa.s}^{-1}$) and 500hPa ( -0.35 $\mathrm{Pa.s}^{-1}$) The distribution of aerosol plume in the whole vertical column (0 to 10 km) can be linked to the instability observed during Event 2.

Since we have discussed on the influence of synoptic conditions on the transport of the aerosol plume toward the SWIO
basin for both events, we have to consider the Amazonian biomass burning activity that occurred during the same time period as Southern African biomass burning activity (Torres et al., 2007) and spread toward the Atlantic Ocean (Figure 6, Figure 7 and Figure 10). This intercontinental transport of Amazonian plume will be discussed in the following section.

### 5.3 Intercontinental transport

To discuss on the intercontinental transport from CAMS associated to South America biomass burning activity, we present the
CO mixing ratio at different levels of pressure: 300, 500 and 700–hPa on September 18 and October 9 (Figure 15). The CO maps at 300–hPa pressure level depict the transport of the South American plumes toward Atlantic Ocean and this transport is more pronounced for during Event 2 (Figure 15a and d). The 500–hPa pressure level shows the mixing between plumes from both continents (Figure 15c and e). This is particularly visible for Event 1, where plumes mix near the Namibian western coast and spread over Southern Africa. CO maps at 700 hPa pressure level look like those observed with IASI on Figure 6a and c,
showing the spread of plumes of each continent. The values CO mixing ratios at 700–hPa pressure level are greater during Event 2 than Event 1, over the Atlantic Ocean and Southern Africa. However, Brazil shows higher values of CO mixing ratio at 700–hPa pressure level during Event 1 than Event 2. This is due to intense fires occurring over this region in September followed by a decrease in October (see the online article News on the on the CAMS, 2022).

Numerous studies have been carried out to study atmospheric aerosols, as well as their regional transport (Ulke et al.,
2007, 2011; De Oliveira et al., 2016; Martins et al., 2018; Cúneo et al., 2022). Works of Ulke et al. (2007, 2011) and Martins et al. (2018) have highlighted the influence of the Southern American low-level jet (SALLJ) on the regional transport of smoke plume over South America. The SALLJ is well described during the Southern American low-level jet Experiment (SALLJEX) and its main role is to transport of moisture from the Amazon to the La Plata Basin (Vera et al., 2006).

Since the TRACE-A experiment, the long-range transport of the South American biomass burning plumes has been poorly
studied. Pickering et al. (1996) and Thompson et al. (1996) have shown that production of ozone in upper troposphere can be associated with detrainment of South American biomass burning emissions by deep convection. Then, these emissions have accumulated from eastern Brazil to Australia. Recently Bencherif et al. (2020) have described the transport of biomass burning





plumes, from South America toward the South Atlantic, that reached Cap Town (in South Africa) in August 2019. The transport was induced by an intensification of a Rossby wave resulting from a sudden stratospheric warming. In contrast with this work,

the South American biomass burning plumes impacted more the Southern Africa and not only the southern region of South Africa.

The propagation of Rossby waves from Pacific Ocean to South America tends to influence the spatiotemporal evolution of the Southern American low-level jet (LLJ, Jones et al., 2023). Jones et al. (2023) have characterized the spatial extent of the LLJ into four types: Central (called also SALLJ), Northern (referred to Orinoco Jet), Andes (correspond to northern and central

branches of the jet) and Peru type. They have shown that the different phases of the Rossby waves train favor the occurrences of Central, Northern or Andes types. The authors also link the influence of ENSO phase to LLJ types. The occurrence of Central types is more frequent in El Niño phase, while Northern type is more frequent in La Niña phase. Lastly, they highlighted that the Madden-Julian Oscillation plays an essential role in creating wave trains that modulate the frequency of SALLJ, particularly the Central type. Since the LLJ is implied in the regional transport of biomass burning plumes and depending on

its configuration, we cannot exclude that this key feature is involved in the long-range transport of South American biomass burning plume during our events. ENSO influences the phase of the Southern Annular Mode (Wang and Cai, 2013) and in combination with the Madden-Julian Oscillation (MJO), they can both influence the configuration of LLJ. From this statement, we cannot exclude the influence of other climate drivers and atmospheric circulation patterns on regional and long-range transport of biomass burning plumes.

**6    Summary and Concluding Remarks**

The present study reports on the spatiotemporal evolution of aerosol optical properties during the BIBAC intensive observational period over two time periods: between September 18 and 23 (Event 1) and between October 9 and 17 (Event 2). These two periods have been established statistically using sun–photometer data at the Skukuza site, by comparing daily observations with the complete dataset. The aerosol classification from sun–photometer is consistent with the one proposed by CALIOP

algorithm, show a predominance of biomass burning aerosols. With the several techniques described in the literature, we have highlighted a predominance of biomass burning aerosols for each event with moderately absorbing optical signatures.

The transport of CO and AOD from ground-based and satellite observations show a north to south and a west to east transport over Southern Africa and toward the SWIO basin. The vertical distribution of aerosols from CALIOP profiles is mainly characterized by elevated smoke spanning mainly from the surface to 6 km. Only one part of Event 2 displays aerosol

layers which extend from the surface to 10 km. These height ranges and aerosol types are consistent with those obtained in the overview presented by Ranaivombola et al. (2023). Those obtained over the western coast of Southern Africa are also consistent with the 2017 study case of Chazette et al. (2019) and Flamant et al. (2022).

A description of the synoptic conditions that have favored the significant perturbation of the aerosols loading from September to October 2022 is provided and discussed in this study. During Event 1, two high-pressure centers in Atlantic Ocean and SWIO

basin have driven the plume by forming the so-called "river of smoke". As reported in the literature (Swap et al., 2003; Flamant



et al., 2022) the meteorological conditions have been influenced by the passage of westerly waves associated with a COL that have favored the eastern transport pathway. Unlike these works, we present a unique result with an evident synoptic condition: the westerly wave has not been converted into a COL during our Event 1. During Event 2, very stable conditions due to the existence of the subtropical anti-cyclonic system accompanied with warm temperatures dominated over the study area. Also,

the southern boundaries of the continent have been crossed by two high-pressure centers. The displacement of these high-pressure centers may have limited the spread of the plume over the continent and its exit into the SWIO basin, as observed during Event 1.

Lastly, we have discussed the long-range transport of biomass burning aerosols from South America that have traveled across the Atlantic Ocean to Southern Africa. The plume has emerged from southern part of Brazil, probably driven by the SALLJ,

which itself is controlled by climate forcing (like ENSO, MJO, etc.) and has propagated over the Atlantic Ocean at higher altitudes than the Southern African plumes. Further works should address the contribution of these plumes to those observed over Southern Africa and the SWIO basin and determine the atmospheric pattern and the climate forcing that have favored the transport between the two continents.

Finally, the "river of smoke" phenomena in the literature and in this work occur during La Niña and since the ENSO phase

does not contribute directly to biomass burning emissions as mentioned by Shikwambana et al. (2022), it would be interesting to make an overview of "rivers of smoke" and analyze their correlation with ENSO, particularly in strong phases like 2010-2011.

*Data availability.* Sun photometer observations Version 3 Level 1.5 are available on https://aeronet.gsfc.nasa.gov/ (last access on April 24 2023). Aerosol MODIS observations "Combined Dark Target and Deep Blue AOD at 0.55 micron for land and ocean, (MOD08_D3_v6.1)", from Aqua platform were downloaded on Giovanni website https://giovanni.gsfc.nasa.gov/giovanni/(last accessed on July 11 2023). Fire

MODIS observations "Thermal Anomalies/Fire locations 1km FIRMS V0061 (MCD14ML)" from Aqua platform were downloaded from the FIRMS platform (https://firms.modaps.eosdis.nasa.gov/, last accessed on November 7, 2023). CALIOP Level-2 Aerosol Profiles (V4-51) were downloaded from the EarthData website (https://search.earthdata.nasa.gov/, last accessed on October 23, 2023). IASI/MetOp-B carbon monoxide (CO) dataset was downloaded from the AERIS platform (https://iasi.aeris-data.fr/co, last accessed on December 8, 2023). CAMS global reanalysis (EC4) were downloaded from the Atmospheric Data Store (ADS) website (https://ads.atmosphere.copernicus.eu/,

last accessed on December 19, 2023)

*Video supplement.* Animations 1 and 2 are available at https://av.tib.eu/media/67052 and at https://av.tib.eu/media/67051, respectively.

*Author contributions.* Conceptualization and methodology, M.R., N.B. and H.B.; software, M.R. and N.B.; data curation, M.R.; writing—original draft preparation, M.R.; writing—review and editing, all co-authors ; visualization, M.R.; funding acquisition, H.B. and N.B. All authors have read and agreed to the published version of the manuscript.



*Competing interests.* The authors declare no conflict of interest.

*Acknowledgements.* We thank the PIs and their staff for establishing and maintaining the Skukuza and UEM Maputo sites used in this investigation. We thank the PHOTONS National Observation Service of the ACTRIS–France infrastructure for the AERONET data and products. We also acknowledge the MODIS and CALIPSO missions scientists and associated NASA personnel for the production of the data used in this research effort. We acknowledge the use of MODIS data from NASA's Fire Information for Resource Management System (FIRMS)
(https://earthdata.nasa.gov/firms), part of NASA's Earth Observing System Data and Information System (EOSDIS). The authors thank to ECMWF-IFS for providing access to CAMS reanalysis. The authors thank Trecia Strydom and Gregor Fieg from the San Parks in the Krüger National Park for their collaboration. This research was jointly funded by the CNRS (Centre National de la Recherche Scientifique) and the NRF (National Research Foundation) in the framework of the IRP ARSAIO and by the South Africa/France PROTEA Program (project No. 42470VA). This study is integrated and supported by the LEFE project BRAAI (Biomass buRning Aerosols over south Africa and Indian
ocean) and the Université de la Réunion through the federation Observatoire des Milieux Naturels et des Changements Globaux (OMNCG) of the OSU-R (Observatoire des Sciences de l'Univers – La Reunion) in the project MOUSSACA (Méthodes et Outils nUmériqueS appliquéeS Aux Composés Atmosphériques) M.R. received a doctoral scholarship from the Reunion Regional Council (Conseil Général de La Réunion).



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



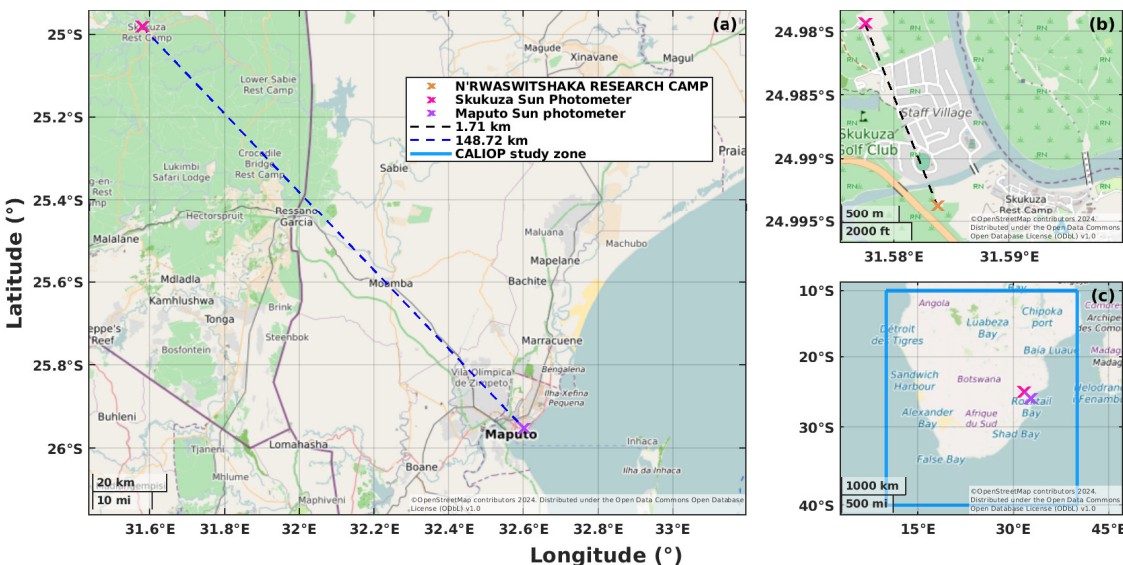

**Figure 1.** Geographical location of the sun–photometer sites and the research camp (panels a and b). Each site is represented by a cross (Skukuza in pink, Maputo in purple and N'Rwaswitshaka Research Camp in orange) and the distance between sites by a dash line (blue between sun–photometers and black between the sun–photometer of Skukuza and the research camp). Panel c represents the CALIOP data selection box.



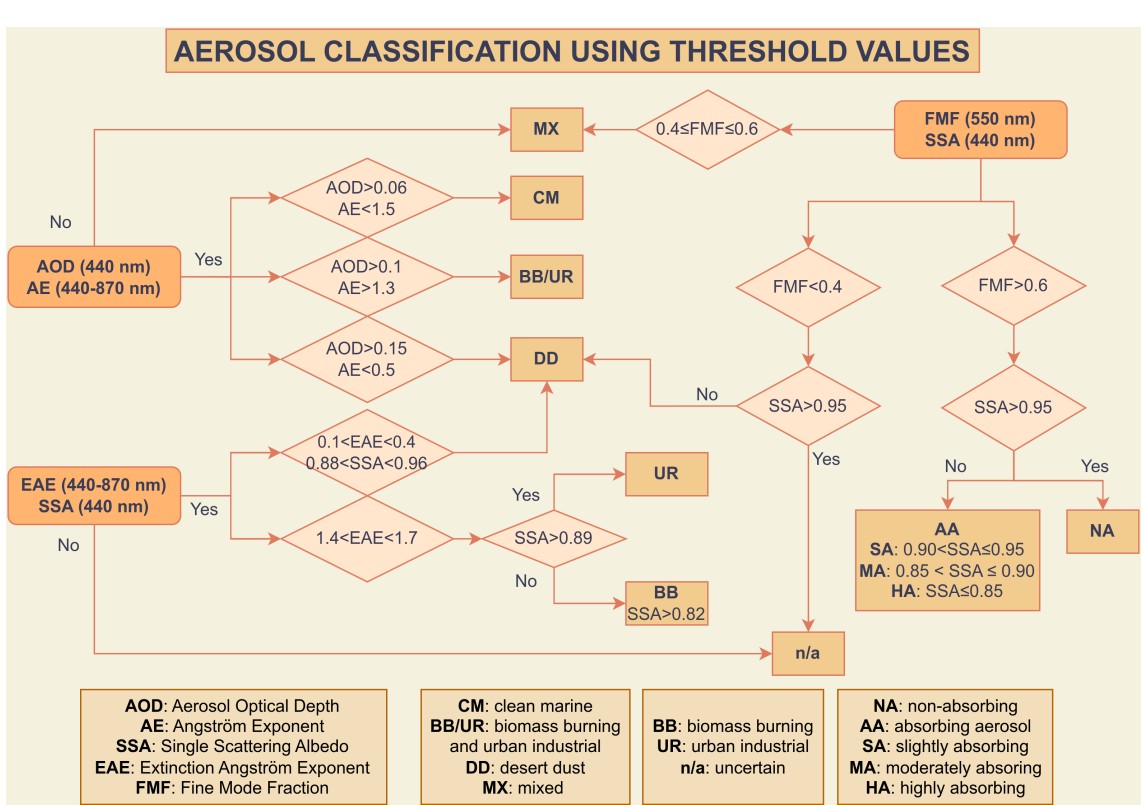

**Figure 2.** A flow chart illustrating the threshold values of aerosol optical parameters used to classify aerosol types.





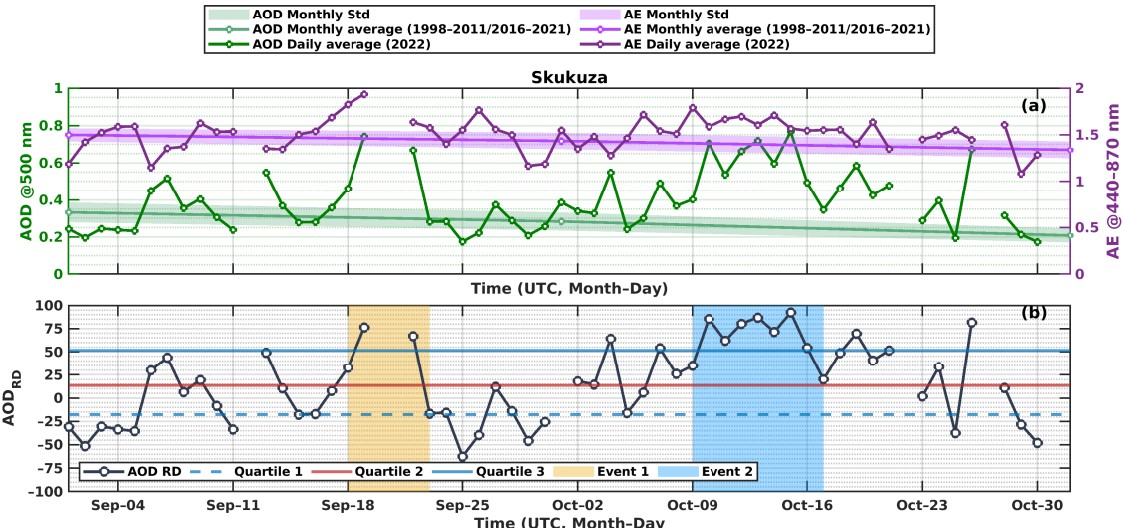

**Figure 3.** Time series of AOD at 500 nm (green lines) and AE at 440-870 nm (purple line) over Skukuza (panel a). Colored lines represent daily 2022 observations, while lighter colored lines depict the multiyear monthly average values for the entire dataset. Shaded areas represent the ±1 standard deviation. Panel b shows the relative difference of AOD between the 2022 and multiyear monthly values (black solid line). The quartiles (Q1, Q2 and Q3) are represented respectively by the blue dashed, red solid lines and blue solid line. The Yellow and blue shaded regions highlight the two selected events (Event 1, September 18 to 23 and Event 2, 9 to 17 October 2022).



**Figure 4.** Aerosol classification from sun–photometer observations at Skukuza during the IOP, with highlights on Event 1 (a-c) and Event 2 (d-f). Relative occurrence frequencies of each aerosol type, in percentage, are based on AOD-AE, EAE-SSA and FMF-SSA methods. Each period is associated with one color: SON in grey, September in dark-blue, event Event-1 in light blue, October in brown and Event 2 in yellow.





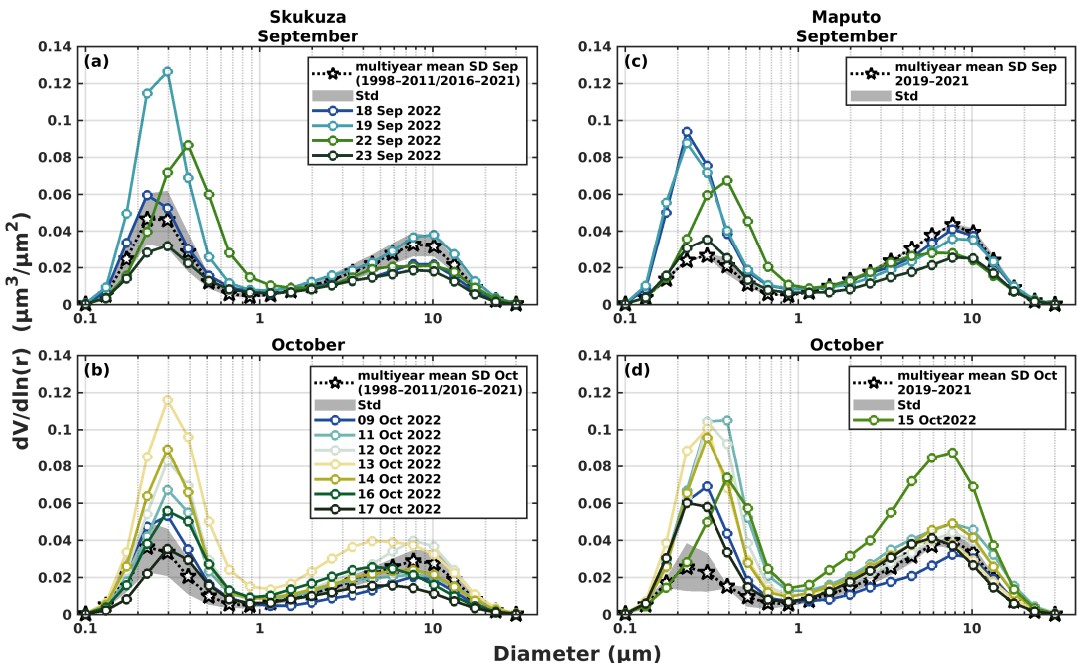

**Figure 5.** Sun photometer aerosol volume size distribution ($\mu m^{-3}/\mu m^{-2}$) at Skukuza (a and b) and Maputo (c and d). Black The black lines (shaded) represent multiyear monthly average ($\pm 1\sigma$) values. Other The other colored lines represent the daily averaged AVSD profiles for each event: the upper panel plots (a and c) correspond to for Event 1, while the lower plots (b and d) and panel b and d for correspond to Event 2. The dates of each AVSD daily profile are listed in the legend of panel a and b, except for non–similar dates between the two sites.





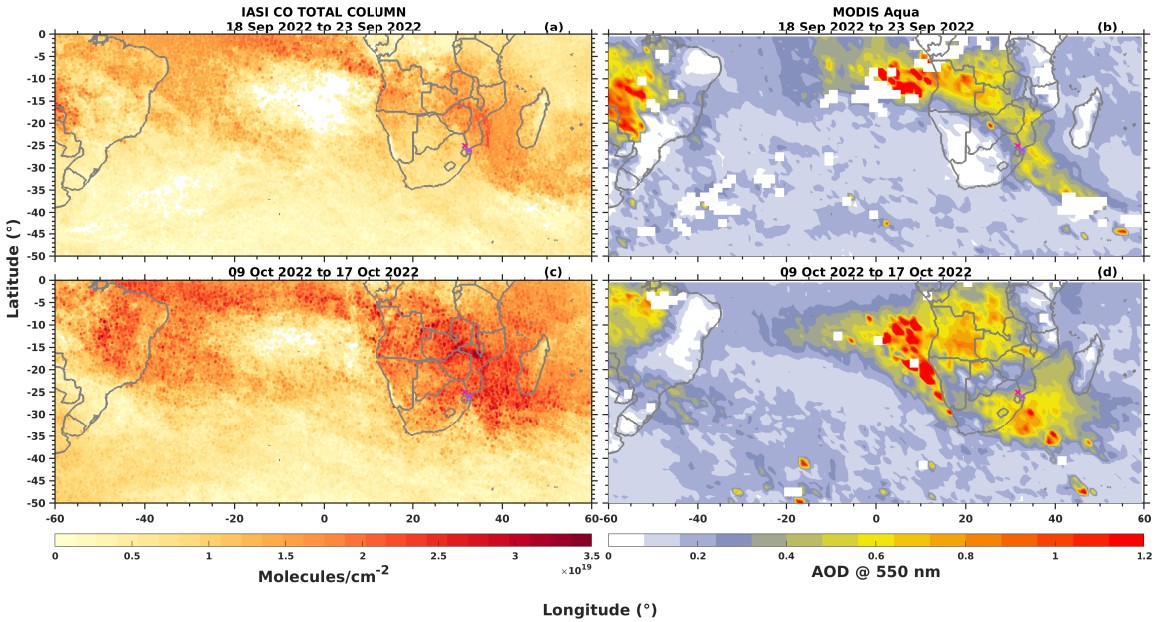

**Figure 6.** Average map of total column of CO (a and c) from IASI and AOD at 550 nm from MODIS-Aqua (b and d) over Southern Africa and East part of South America. Panels (a) and (b) correspond to an average over Event 1, while panels (c) and (d) for Event 2. Note that the blank on panels b and d could correspond to cloud contamination.

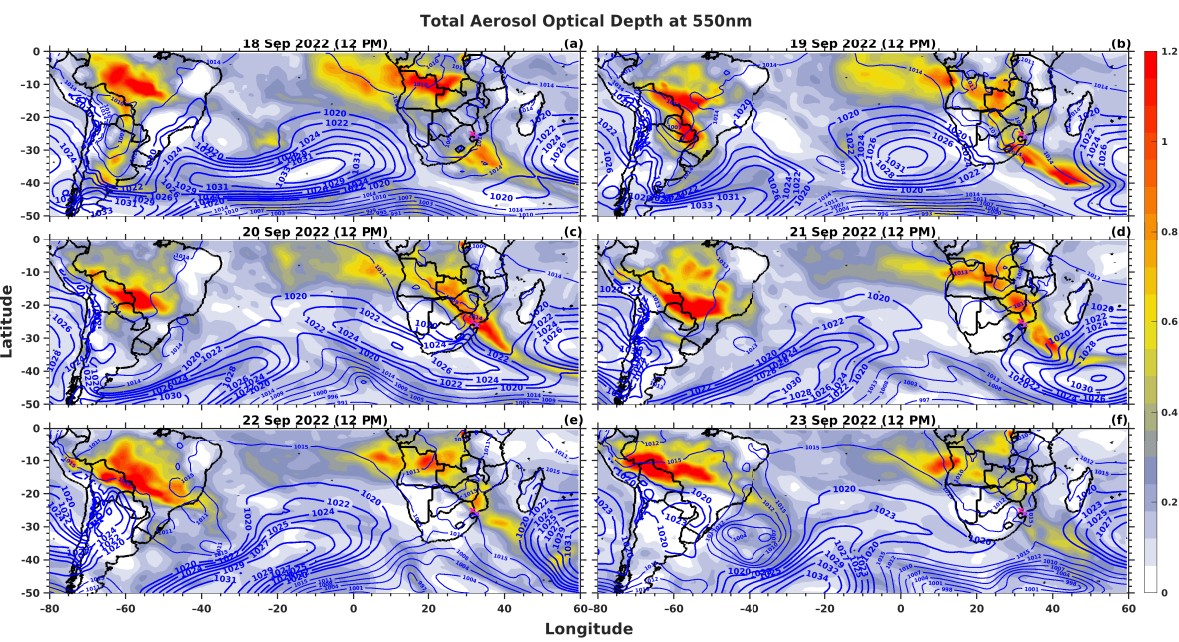

**Figure 7.** Daily map of AOD at 550 nm (color shading) and mean sea level pressure in hPa (blue isolines) at 12:00 UTC from CAMS, successively from (a) to (f) on 18 to 23 September 2022, over Southern Africa, Atlantic Ocean, and South America during Event 1



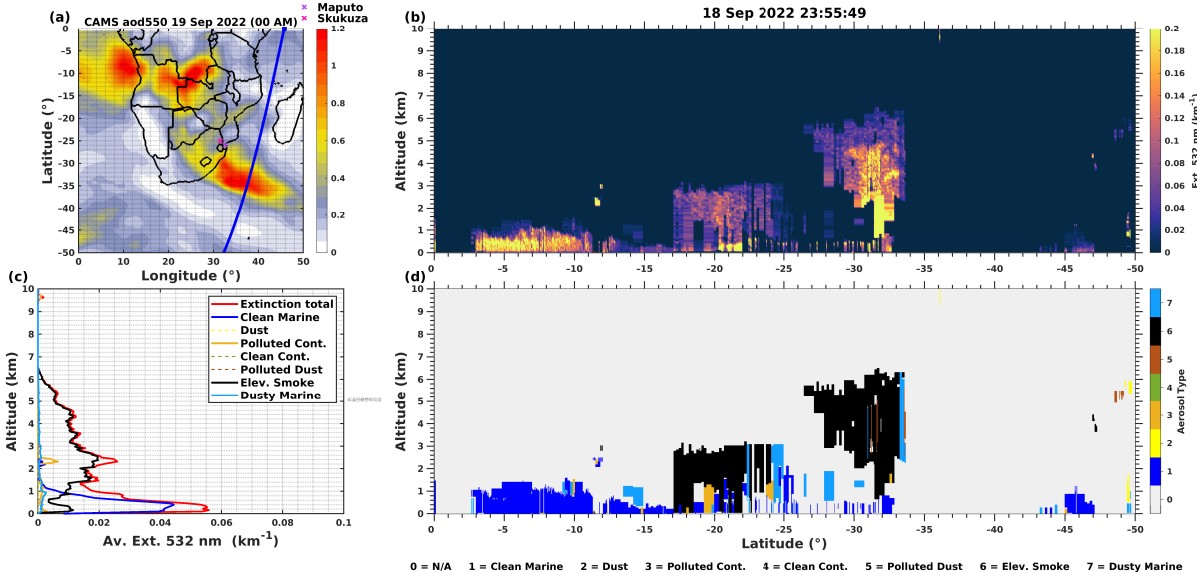

**Figure 8.** CALIOP observations and CAMS Reanalysis on September 18 at 23:55 UTC and September 19 at 00:00 UTC. CALIPSO overpass (blue line) and CAMS AOD at 550 nm are in panel a. Cross-section of extinction coefficient at 532 nm and aerosol type: clean marine (dark-blue line); dust (yellow line); polluted continental/smoke (orange line); clean continental (green line); polluted dust (brown line); elevated smoke (black line); and dusty marine (sky-blue line) are in panel b and d, respectively. Vertical profile of extinction total (in red) and for each aerosol type (remaining colors) are in panel c.



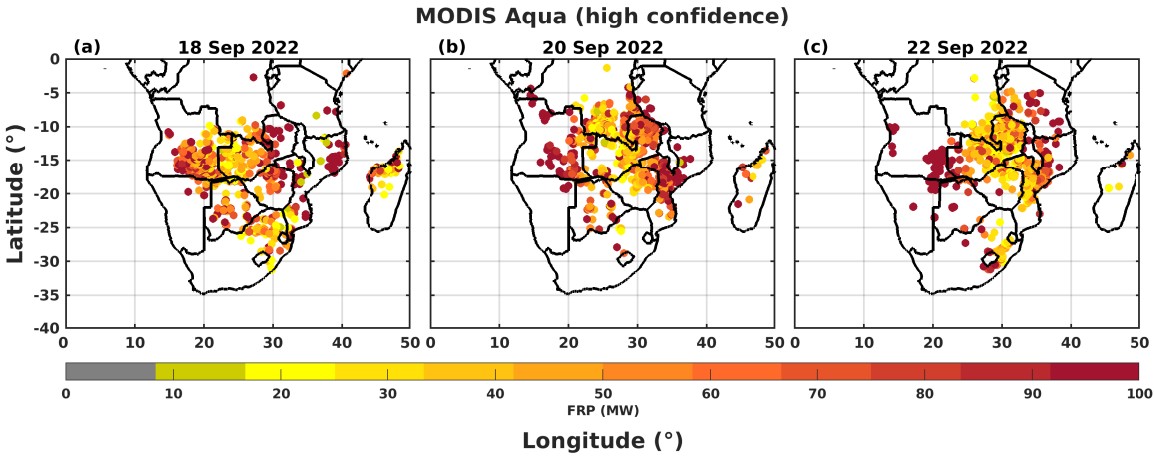

**Figure 9.** Fire radiative power (FRP in MW) over Southern Africa on September 18, 20 and 22.



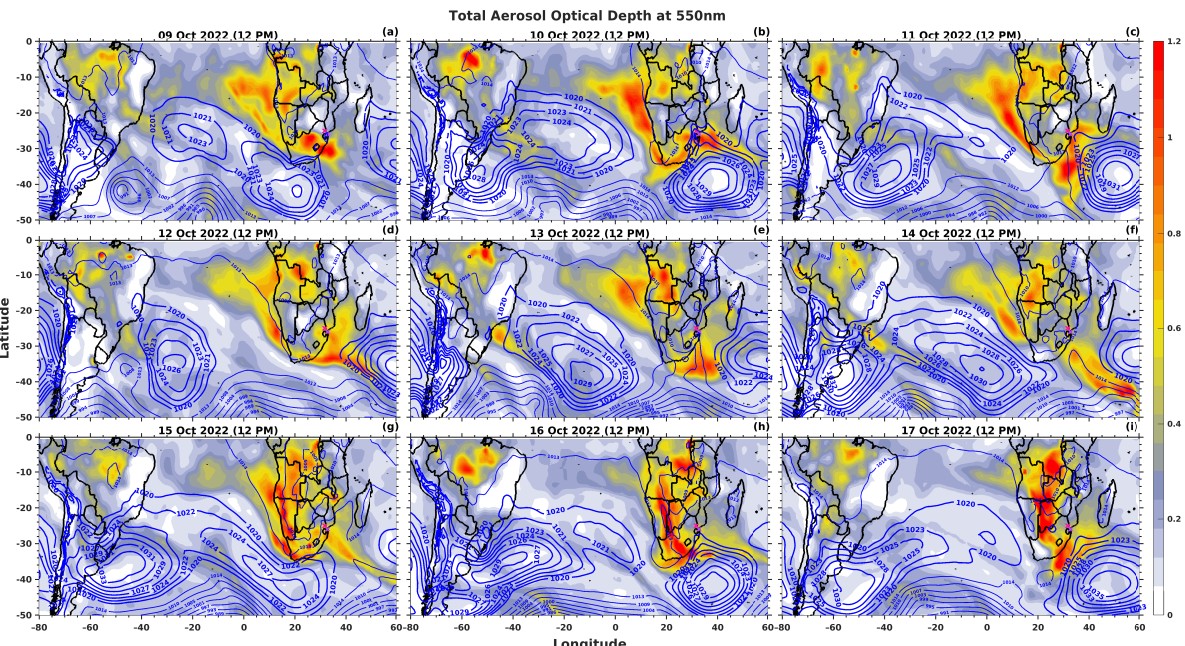

**Figure 10.** Same as Figure 7 but for Event 2.



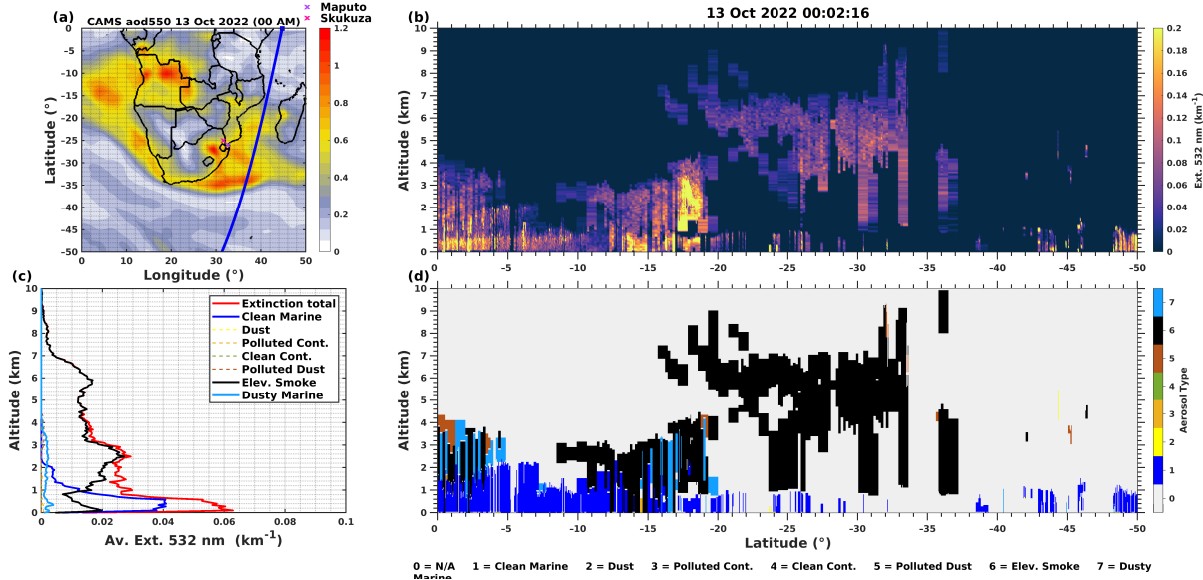

**Figure 11.** CALIPSO observations and CAMS Reanalysis on October 13 at 00:02 UTC and at 00:00 UTC, same as for Figure 8.



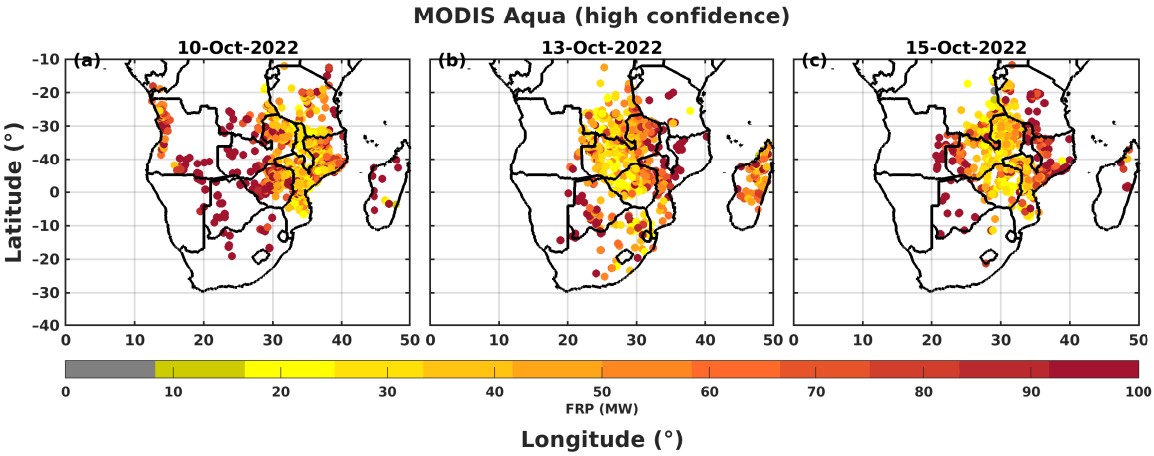

**Figure 12.** Fire radiative power (FRP in MW) over Southern Africa on October 10, 13 and 15.

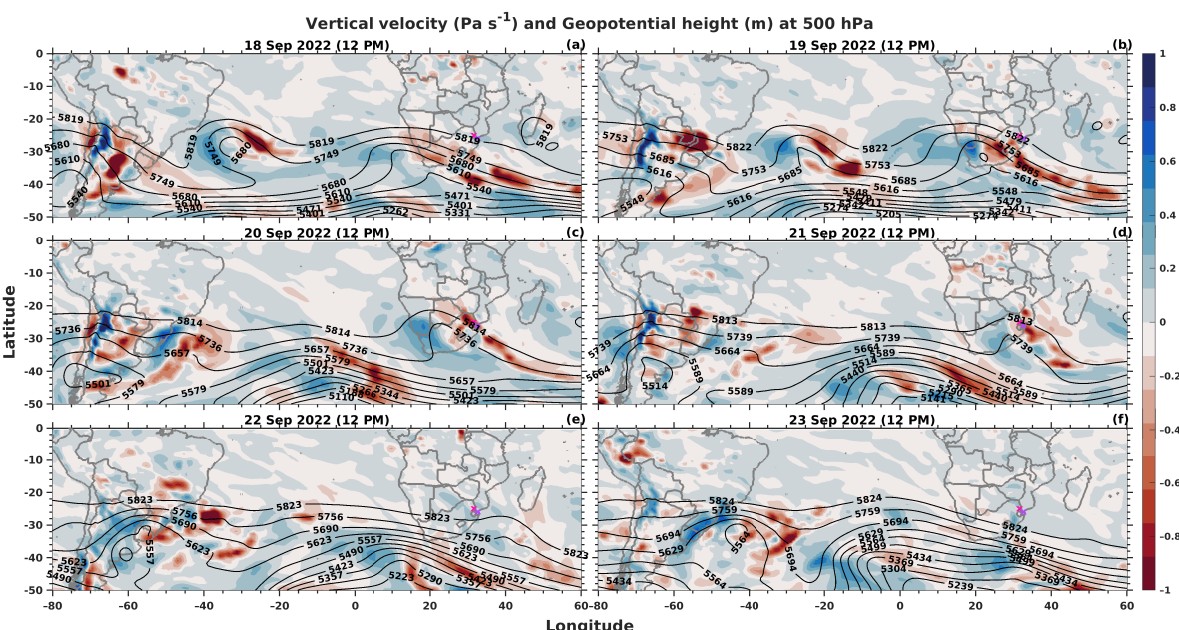

**Figure 13.** Daily map of vertical velocity $(\mathrm{Pa.s}^{-1})$ at 700 hPa (color shading) and geopotential height (m) at 700 hPa (black isolines) at 12:00 UTC, successively from (a) to (f) on 18 to 23 September 2022, over Southern Africa, Atlantic Ocean, and South America during Event 1. Negative values of vertical velocity are associated with upward motion, while positive to downward motion.



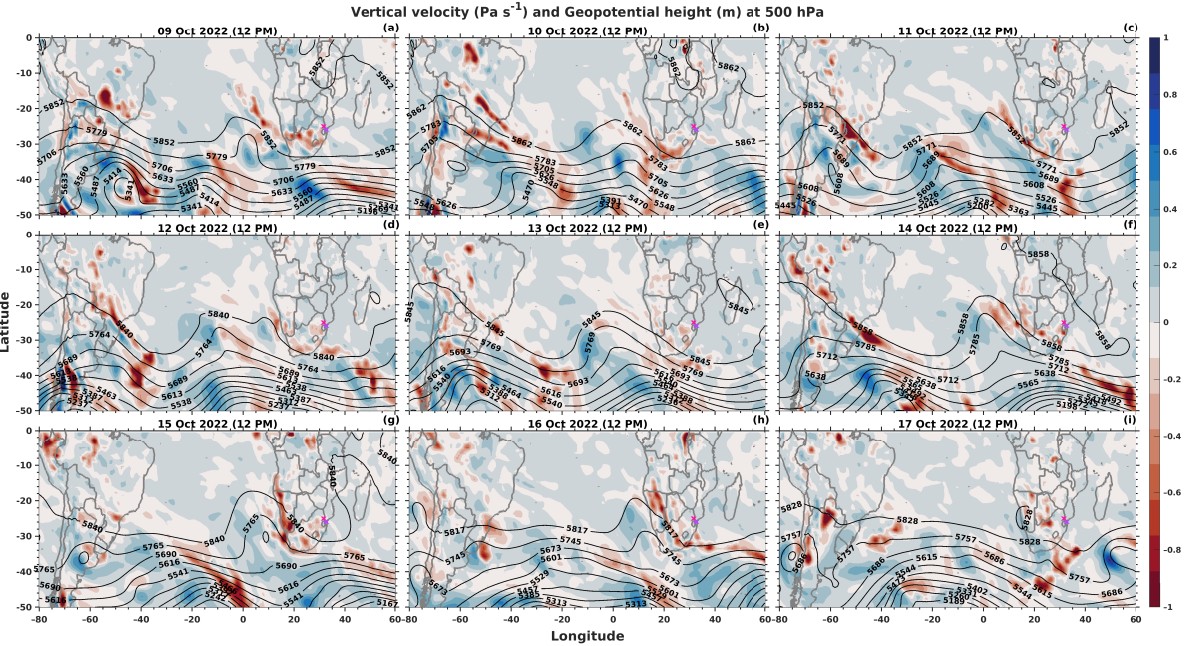

**Figure 14.** Same as for Figure 13 but for the Event 2.



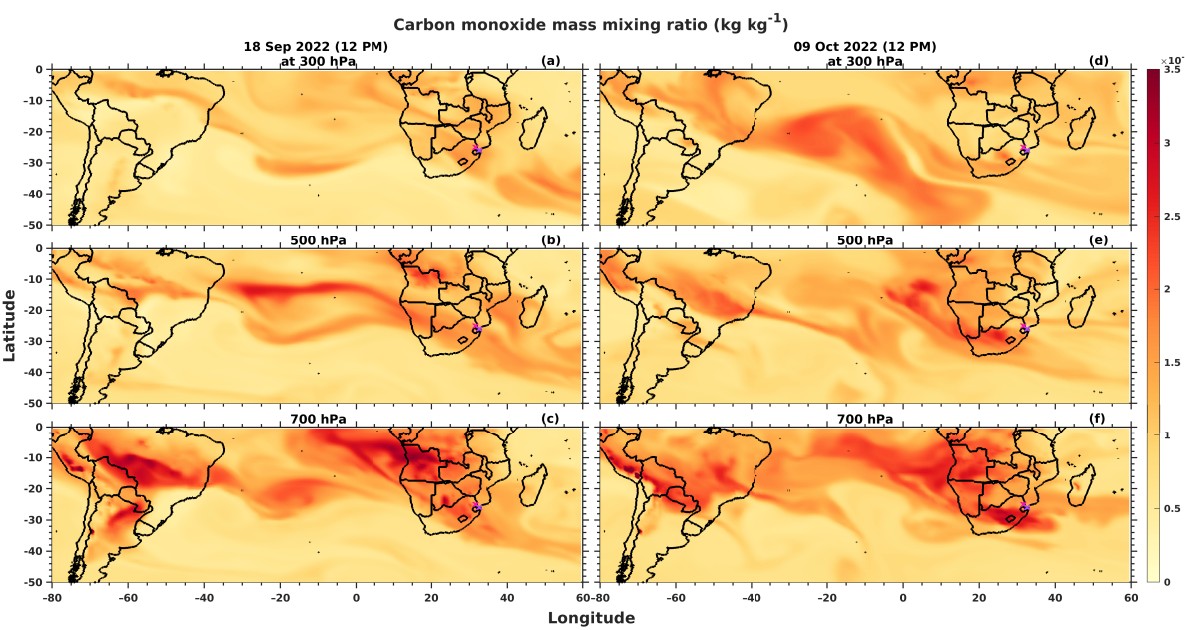

**Figure 15.** Daily map of mixing ratio of CO (color shading) at 300, 500 and 700 hPa (from top to bottom) over Southern Africa, Atlantic Ocean, and South America on September 18 (panels a to c) and October 9 (panels c to f) at 12 PM.



**Table 1.** Summary of Relative occurrence frequencies of predominant aerosol type from sun–photometer observations at Skukuza during the
IOP, during Event 1 and 2. Relative occurrence frequencies are in percentage and based on AOD-AE, EAE-SSA and FMF-SSA methods.

|  | Event 1 | Event 2 |
|---|---|---|
| **AOD-AE** | | |
| **BB/UR** | 100 | 100 |
| **EAE-SSA** | | |
| **BB** | 83.3 | 50.0 |
| **UR** | 16.6 | 50.0 |
| **FMF-SSA** | | |
| **NA** | 0.83 | 2.78 |
| **AA** | 100 | 97.22 |
| **SA** | 14.29 | 38.89 |
| **MA** | 80.95 | 58.33 |
| **HA** | 4.76 | 0 |