# Peer review of "Characterization of AOD anomalies in September and October 2022 over Skukuza in South Africa"

_EGUsphere, 2024_

## Referee Comment (RC2)

This submitted manuscript analyse aerosol properties during two large biomass burning events using data from multiple sources, including the ground-based networks, satellite remote sensing, and reanalyses over South Africa. With these data, the authors confirmed two dust events during Sep and Oct, 2002 based on the classification results. All subsequent analyses on the spatiotemporal distribution of AOD and CO, the vertical profile of aerosol extinction coefficient are conducted on a daily basis.

The work is within scope for the journal and there is new material for this publication to be warranted. As the authors claimed, few such studies have been conducted in South Africa, which may attract readers' interest and contribute to the aerosol community. Additionally, the figures are nice and informative. There are, however, some important missing aspects and points that need clarification. This submitted manuscript is not well prepared, e.g., I can find some abbreviations are not defined or defined at a wrong place. Another big issue that needs to be addressed is the language. I'm aware that the authors are not native English speakers, while it is extremely important to ensure the language is polished for successful publication in this prestigious journal. Please pay attention to the language in your revised manuscript, I recommend major revisions and would like to review the next version.

Abstract. I recommend the authors to follow a more typical structure, including the general background of the study, the aim, the objectives, the novelty, and the main findings in the results. The current form is not mature for a renowned journal like ACP. Additionally, not every abbreviation is fully defined, e.g., what are CALIOP, SWIO, and COL? Only mentioned without definitions.

Line 17, Page 1. What does "their effects on radiative forcing" mean? The scattering and absorbing effect and the indirect effect on cloud microphysics of aerosols all affect atmospheric radiation budget. This sentence should be checked.

Line 19, Page1. I cannot agree with the authors on that the lack of aerosol observations is the reason for the more importance of assessing climate change. Rather, it can be the reason for that the related assessment work is difficult.

Line 22, Page2. "biomass burning activities" should be plural.

Line 25 Page 2. Again, the authors should make sure the submission is well prepared. "AOD" should be defined when it appears for the first time.

Line 27, Page 2. "low number of observations" of what? Aerosols or greenhouse gases?

Line 65, Page 3. Did SAFARI-92, SAFARI-2000 and AEROCLO-sA observe vertical profiles for aerosols?

Figure 1. Only three sites involved in this study. I recommend the authors to label all their names in each panel.

Line 104, Page 4. AERONET measurements are not only made during daytime.

Line 106, Page 4. What's the difference between AERONET data at Level 1.5 and Level 2.0 ?

Lines 107-108, Page 4. Spectral channel information should be provided for SSA, EAE, and FMF. "Ångström Exponent (AE) at 440nm and 870nm" is not clear. This should be confirmed.

Lines 111 to 113, Page 4. The two citations "Eck et al., 1999" and "Dubovik et al., 2006" were published many years ago. Giles et al., (2019, https://doi.org/10.5194/amt-12-169-2019) is a more appropriate study for AERONET AOD validation for the latest V3 product.

Lines 114 to 118, Page 4. The latest AERONET V3 database has been released for several years and related articles about its accuracy have also been published. Why authors only cited papers for the former version of AERONET data.

Line 123, Page 5. What does "all-point" data mean?

Lines 124 to 142, Page 5. I understand the authors want to highlight how the selected parameters classify aerosol types. As the authors said, they set many thresholds for different parameters but they didn't explain how and why they set such values and the references. So, I recommend rewrite this paragraph to focus more on the current study and interpreting figure 2.

Line 140, Page 5. Where is the equation 2? If the authors cite this equation in another study, they should give this equation and cite the study.

Line 150, Page 5. Not clear. My understanding is that MODIS is only carried on Aqua.

Line 156, Page 5. "MOD08_D3_v6.1" is not correct. The MODIS/Aqua product is named as "MYD08_D3".

Line 156, Page 5. The study uses the MODIS AOD product, but the authors give much information of MODIS bands and original spatial resolution for raw data. I recommend the authors to include more information on the MODIS AOD product, e.g., its spatial resolution, accuracy compared to AEROENT data, quality control flags, etc.

Line 186, Page 7. Please refer to my comment on MODIS data. Why did you select the IASI/MetOp-B CO dataset? Because of high accuracy or spatial data coverage?

Line 199, Page 7. Why mslp is not capitalized?

Line 200, Page 7. The units within brackets should be formatted using a different font style from the surrounding text.

Line 284, Page 10. Though AOD shows similar patterns with CO, it cannot be easily considered that AOD is the result of biomass burning. Besides, differences are large during the event 2.

Line 339, Page 11. I might miss something. How do biomass burning activities increase the aerosol radius?

Lines 369 to 370, Page 12. The numbers do not correspond to what they should indicate.

Line 435, Page 14. Figure 13?

---

## Author Response (AR2)

**RESPONSE TO EDITOR**

**Dear Authors,**

**Thanks for your efforts to improve your manuscript during the review process. Thank you to the reviewers who's comments and suggestion have been a great and good contribution. As the review is approaching its end, please make sure your abstract respect the ACP guidelines limitation to 250 words. It is currently more than 300 words and I believe it can be reduced and fit the limitation while keeping a very good quality. Best**

**Authors:**

**Dear Editor,**

**Thank you for your message and for your guidance throughout the review process. Following your request, we have revised the abstract to comply with the ACP guidelines, reducing it to 250 words while maintaining its clarity and quality. Please find the updated abstract in the revised manuscript.**